# Antimicrobial Peptides Active in In Vitro Models of Endodontic Bacterial Infections Modulate Inflammation in Human Cardiac Fibroblasts

**DOI:** 10.3390/pharmaceutics14102081

**Published:** 2022-09-29

**Authors:** Giulia Marianantoni, Giada Meogrossi, Eva Tollapi, Alessandro Rencinai, Jlenia Brunetti, Crystal Marruganti, Carlo Gaeta, Alessandro Pini, Luisa Bracci, Marco Ferrari, Simone Grandini, Chiara Falciani

**Affiliations:** 1Department of Medical Biotechnologies, University of Siena, 53100 Siena, Italy; 2School of Dental Medicine, University of Siena, 53100 Siena, Italy; 3Setlance srl, 53100 Siena, Italy; 4Laboratory of Clinical Pathology, Siena University Hospital, 53100 Siena, Italy

**Keywords:** oral bacterial infections, antimicrobial peptides, anti-inflammatory activity, cardiovascular diseases

## Abstract

Endodontic and periodontal disease are conditions of infectious origin that can lead to tooth loss or develop into systemic hyperinflammation, which may be associated with a wide variety of diseases, including cardiovascular. Endodontic and periodontal treatment often relies on antibiotics. Since new antimicrobial resistances are a major threat, the use of standard antibiotics is not recommended when the infection is only local. Antimicrobial peptides were recently demonstrated to be valid alternatives for dental treatments. The antimicrobial peptide M33D is a tetrabranched peptide active against Gram-negative and Gram-positive bacteria. It has a long life, unusual for peptides, because its branched form provides resistance to proteases. Here the efficacy of M33D and of its analog M33i/l as antibiotics for local use in dentistry was evaluated. M33D and M33i/l were active against reference strains and multidrug-resistant clinical isolates of Gram-negative and Gram-positive species. Their minimum inhibitory concentration against different strains of dental interest was between 0.4 and 6.0 μM. Both peptides acted rapidly on bacteria, impairing membrane function. They also disrupted biofilm effectively. Disinfection of the root canal is crucial for endodontic treatments. M33D and M33i/l reduced *E. faecalis* colonies to one-twentieth in a dentin slices model reproducing root canal irrigation. They both captured and neutralized lipopolysaccharide (LPS), a bacterial toxin responsible for inflammation. The release of IL-1β and TNFα by LPS-stimulated murine macrophages was reduced by both peptides. Human cardiac fibroblasts respond to different insults with the release of proinflammatory cytokines, and consequently, they are considered directly involved in atherogenic cardiovascular processes, including those triggered by infections. The presence of M33D and M33i/l at MIC concentration reduced IL6 release from LPS- stimulated human cardiac fibroblasts, hence proving to be promising in preventing bacteria-induced atherogenesis. The two peptides showed low toxicity to mammalian cells, with an EC50 one order of magnitude higher than the average MIC and low hemolytic activity. The development of antimicrobial peptides for dental irrigations and medication is a very promising new field of research that will provide tools to fight dental infections and their severe consequences, while at the same time protecting standard antibiotics from new outbreaks of antimicrobial resistance.

## 1. Introduction

Antimicrobial resistance is a major threat to human health. It is responsible for an estimated 33,000 deaths per year in the EU [1,2] and comes with a heavy economic burden. The lack of effective new antimicrobials is due to little investment in antibiotic research and to excessive use and misuse of antibiotics to treat humans and animals. Trivial and life-threatening infections are often treated with the same antibiotics, generating an increasing number of resistant strains that can easily turn an unimportant bacterial infection into a severe disease.

Endodontic and periodontal diseases begin with bacterial colonization. Though initially local, chronic infection may become a source of antimicrobial-resistant bacteria. Chronic disease also triggers sustained release of proinflammatory mediators that are essential to the development of inflammation [3,4] and can eventually have detrimental effects on distant organs. Many epidemiological studies suggest that periodontitis is associated with atherosclerotic cardiovascular disease [5,6,7]. Bacteria enter the systemic circulation, stimulating atherogenesis through endothelial damage and inflammation, as shown by the fact that experimental bacteremia induced by *P. gingivalis* in animals leads to atherogenesis [8]. Furthermore, patients with periodontitis constantly produce proinflammatory cytokines that can reach the liver and trigger release of acute-phase response proteins (C-reactive protein, alpha 1-acid glycoprotein, fibrinogen, and serum amyloid A), causing endothelial damage and triggering atherogenesis [9].

The human oral microbiota is known to contain as many as 700 microorganisms. The average person has about 250 species. They include Gram-negative bacteria, such as *Escherichia coli*, *Porphyromonas gingivalis*, and *Pseudomonas aeruginosa*, and Gram-positive bacteria, such as *Actinomycetes*, *Enterococci*, *Lactobacilli*, *Staphylococci*, and *Streptococci*, with *Streptococcus mutans*, which causes dental caries, and fungi, mainly *Candida*. Among the microorganisms causing periodontitis, Enterococci are major players [10].

Enterococci are Gram-positive colonizers of the mouth and mammalian gastrointestinal tract and normally live in healthy association as commensals in humans and animals [11]. However, *Enterococcus faecalis* can cause bacteremia and severe infections such as meningitis and endocarditis [12]. In the oral cavity, *E. faecalis* has been detected primarily in the root canals of patients with post-treatment apical endodontitis or refractory apical periodontitis, suggesting a causal role in the progression of these diseases [13,14]. The emergence of antibiotic-resistant *E. faecalis*, such as vancomycin-resistant *E. faecalis* (a glycopeptide-resistant strain), prompted research and development of new antibiotics.

In dentistry, antimicrobial peptides have been suggested to fight pathogens in implants and dental adhesives [15]. The peptide M33D is a tetrabranched peptide active against Gram-negative and Gram-positive bacteria [16,17]. It has a long life, unusual for peptides, since the branched form imparts resistance to plasma, serum, and bacteria proteases [18].

M33D shows MICs of 0.7–6.0 μM (4–32 µg/mL) against multiresistant pathogens of clinical interest, such as Gram-positive *Staphylococcus aureus*, *Staphylococcus saprophyticus*, and *Enterococcus faecalis*, including a vancomycin-resistant strain, and various Gram-negative enterobacteriaceae [17]. The activity is accompanied by a low frequency of resistance selection [17].

M33D has been successfully tested against multidrug-resistant strains of seriously dangerous Gram-negative species, such as *Escherichia coli, Klebsiella pneumonia, Acinetobacter baumannii*, and *Pseudomonas aeruginosa*, resistant to carbapenem, extended-spectrum cephalosporin, fluoroquinolone, aminoglycoside, and, importantly, colistin [17].

M33D also neutralizes lipopolysaccharide and lipoteichoic acid, thus exerting anti-inflammatory activity by reducing expression of cytokines, enzymes, and transcription factors (TNF-α, IL6, COX-2, KC, MIP-1, IP10, iNOS, NF-kB) involved in the onset and evolution of inflammatory processes [17].

By virtue of all these characteristics and its low toxicity [17], M33D and its analog M33i/l are considered good candidates to fight endodontic and periodontal disease.

## 2. Materials and Methods

### 2.1. Peptide Synthesis

The peptides M33D (kkirvrlsa)_4_K_2_KβA-OH and M33i/l (kklrvrlsa)_4_K_2_KβA-OH were solid-phase synthesized by standard Fmoc chemistry using D-amino acids with a Syro multiple-peptide synthesizer (MultiSynTech, Witten, Germany). We used a TentaGel-PHB 4 branch βAla Wang-type resin (Rapp Polymere, Germany), which carries the branching core in L-form, Fmoc_4_-Lys_2_-Lys-β-Ala, as previously described [16]. The linear homolog (KKIRVRLSA) was synthesized on TentaGel-PHB Wang-type resin (Rapp Polymere, Germany) following the same procedure. Sidechain-protecting groups were 2,2,4,6,7-pentamethyldihydrobenzofuran-5-sulfonyl for R, t-butoxycarbonyl for K, and t-butyl for S. The final product was cleaved from the solid support, deprotected by treatment with TFA containing triisopropylsilane and water (95/2.5/2.5), and precipitated with diethyl ether. Crude peptide was purified by reversed-phase chromatography on a Phenomenex Jupiter C18 column (300 Å, 10 mm, 250, 610 mm), using 0.1% TFA/water as eluent A and methanol as eluent B, in a linear gradient from 80% A to 20% A in 30 min. Final peptide purity and identity were confirmed by reversed-phase chromatography on a Phenomenex Jupiter C18 analytical column (300 Å, 5 mm), M33D RT = 21 min, M33i/l RT = 22 min, and by mass spectrometry with a Bruker Daltonics Ultraflex MALDI TOF/TOF), M33D M+(found) = 4682.36, M33i/l M+(found) = 4682.86.

### 2.2. Human Serum Protease Stability

A pool of sera from healthy volunteers (n = 5) was diluted to 50% with RPMI 1640 medium. A 5 μg/mL solution of the peptides in 50% serum was incubated at 37 °C for different time intervals (2, 4, 24, 48, and 72 h). At the established time, trichloroacetic acid was added at a 7.5% concentration, and the sample was centrifuged at 12,000–13,000× *g* for 15 min. Part of the supernatant was used for the MALDI-TOF analysis (2 μL) and part (250 μL) for the HPLC analysis, after being diluted with 1% TFA (750 μL). Time-zero HPLC and MS-spectroscopy spectra were obtained immediately after mixing the peptide with 50% serum. The presence of the intact peptide was determined in HPLC by spotting and integrating the peptide peak at the correct retention time. The identity of the peptide peak was further also confirmed by MALDI mass spectrometry.

### 2.3. Peptide Structure Prediction

De novo prediction of peptide tertiary structure was achieved with APPTEST [19], a computational protocol that combines the predictive power of neural networks with structural biology software programs XPLOR-NIH and CYANA. Neural networks of APPTEST were formed on experimental model structures derived from the Protein Data Bank (PDB, https://www.rcsb.org/, accessed on 23 May 2022) to predict structural constraints, which are used in restrained molecular dynamics simulations to produce a final set of structures.

### 2.4. Minimum Inhibitory Concentrations

The bacteria reported in Table 1 were used for conventional susceptibility testing experiments. M33D and M33i/l MICs were determined in triplicate on reference and clinical strains using a microdilution assay, performed according to the guidelines of the Clinical and Laboratory Standards Institute [20]. Briefly, strains were grown on Mueller–Hinton (MH) agar plates (Becton Dickinson, Franklin Lakes, NJ, USA) and a single colony for each strain was picked using a sterile cotton swab, streaked in sterile cation-supplemented MH, and measured with a densitometer (Densichek, bioMèrieux, Marcy l’Etoile, France) up to 0.15 optical density. Wells of a microtiter plate containing 50 μL serial-doubling dilutions of the peptides were inoculated with an equal volume of bacterial suspension. The final bacterial inoculum was 5 × 10^4^ CFU/well in a volume of 100 μL. MIC values were recorded after incubation of plates at 35° for 18–20 h. Assays were performed in triplicate, and the median MIC values were recorded.

### 2.5. Bacterial Membrane Interaction

Mid-log phase *E. faecalis* PE or *E. coli* LC711, resuspended at 1 x 10^8^ CFU/mL in PBS-glucose, was incubated in an orbital shaker at 37 °C for 15 min. Then PI or SYTOx green dye was added to final concentrations of 5 μg/mL and 5 μM, respectively. The suspension was vortexed, and 200 μL was added to the wells of a black 96-well plate (Optiplate). Samples were preincubated at 37 °C with fluorescence measurements every minute for 5 min, or until readings stabilized. The plate was then ejected. Peptides (2× and 4 × MIC) were added in duplicate wells. The plate was immediately returned to the reader to continue monitoring PI and SYTOx green (PI; λex = 535, λem = 617 nm; SYTOx green λex = 504, λem = 523 nm), every 1 min for 100–120 min.

### 2.6. Antibiofilm Activity

Biofilm inhibition and destruction activity were evaluated by two independent parameters: the biofilm prevention concentration (BPC) and the minimal biofilm inhibition concentration (MBIC) [21,22]. Briefly, for BPC, an overnight culture of *E. faecalis* PE, *E. faecalis* 51299, *E. coli* TG1, or *E. coli* LC711 bacteria was grown to 0.8 optical density (OD) in Tryptic Soy Broth (TSB) 0.25% glucose. This bacterial suspension (100 μL) was added to wells in triplicate in a 96-well U-bottom plate, then 100 μL M33D and M33i/l was added to the well with twofold serial dilution of each peptide.

**Table 1 pharmaceutics-14-02081-t001:** MIC of M33d and M33i/l against bacterial strains representative of various pathogenic species, including MDR strains of clinical origin.

Strains	MIC μM (μg/mL)		Relevant Features
	M33D	M33i/l	
*P. aeruginosa ATCC 27853*	1.5 (8) *	0.8 (4)	
*P. aeruginosa AV 65*	3 (16) *	1.5 (8)	FQ^r^ AG^r^ ESC^r^ NEM^r^ (MBL/IMP-13)
*K. pneumoniae ATCC 13833*	3 (16) *	3 (16)	
*K. pneumoniae 7086042*	6 (32) *	6 (32)	FQ^r^ AG^r^ ESC^r^ NEM^r^ (MBL/VIM-1)
*E. coli ATCC 25922*	3 (16) *	3 (16)	
*E. coli TG1*	1.5 (8)	1.5 (8)	
*E. coli W03BG0025*	3 (16) *	3 (16)	FQ^r^ AG^r^ ESC^r^ (ESBL/CTX-M-15)
*E. coli LC711*	1.5 (8)	0.3 (1.6)	ColR
*A. baumannii RUH 875*	3 (16) **		
*S. aureus USA 300*	1.5 (8) *	1.5 (8)	MR
*S. aureus 3851*	0.8 (4) *	0.8 (4)	MR VAN ^i^
*S. epidermidis ATCC 14990*	0.4 (2) *	0.8 (4)	
*S. epidermidis 6154*	0.8 (4) *	0.8 (4)	MR
*E. faecalis ATCC* 51299	3 (16)	1.5 (8)	
*E. faecalis* PE ^a^	1.5 (8)	1.5 (8)	Tetracycline-resistance ^b^
*E. faecalis Fi74B1*	0.8 (4)		
*E. faecalis Fi74B4*	1.5 (8)		
*E. faecium FI81B1*	0.8 (4) **		GLY
*C. striatum ATCC*	1.5 (8)		
*S. pyogenes* CCUG2557	6 (32)		
*S. pneumoniae* ATCC 49619	6 (32)		
*C. albicans* ATCC 90028	5 (24)		
*C. glabrata* ATCC 90030	12 (64)		
*C. krusei* ATCC 6268	6 (32)		

The strains tested included reference strains (indicated) and clinical isolates (mostly with an MDR phenotype). Relevant resistance traits and resistance mechanisms are indicated. FQ^r^, resistant to fluoroquinolones; AG^r^, resistant to amino-glycosides (gentamicin, amikacin and/or tobramycin); ESC^r^ resistant to expanded-spectrum cephalosporins (through ESBL, extended-spectrum-β-lactamase type CTX-M-15); NEM^r^, resistant to carbapenems (imipenem and/or meropenem) (through MBL metal-β-lactamase type IMP-13 or VIM-1); ColR, resistant to colistin/polymyxin E [23]; MR, methicillin-resistant; VAN ^i^, vancomycin-intermediate; GLY, glycopeptide resistant. ^a^ Post-endodontic treatment isolate. ^b^ Determined with standard antiobiogram. * values from [16]. ** values from [17].

Untreated bacteria were used as positive control. After 24 h of static incubation at 37 °C, wells were washed thrice with PBS and fixed with PFA 4% in PBS (200 μL) for 30 min at room temperature. Wells were then washed 3 times, and crystal violet (1% in water) was added and incubated for 30 min in the dark. Wells were again washed, and the color dissolved with ethanol/acetone 80:20. Color was read at 595 nm. The BPC is the lowest concentration of an antibiotic that results in an OD595 nm difference of ≤10% in the mean of two control well readings.

For MBIC, the bacterial suspension (200 μL) of *E. faecalis* PE, *E. faecalis* 51299, *E. coli* TG1, or *E. coli* LC711, obtained as above, was added to wells in triplicate, in a 96-well U-bottom plate and incubated at 37 °C for 24h. The supernatant was removed, and 200 μL M33D and M33i/l diluted in TSB was added to the wells in twofold serial dilutions of each peptide.

Untreated bacteria were used as positive control. After 14 h of static incubation at 37 °C, wells were washed thrice with PBS and fixed with PFA 4% in PBS (200 μL) for 30 min at room temperature. Wells were then washed three times, and crystal violet (1% in water) was added and incubated for 30 min in the dark. Wells were again washed, and the color dissolved with ethanol/acetone 80:20. Color was read at 595 nm. The MBIC is the lowest concentration of an antibiotic that results in an OD595 nm difference of ≤10% of the mean of two control well readings.

### 2.7. Dentin Slice Model

Extracted human molars and premolars were obtained from patients according to a protocol approved by the Clinical Ethics Committee of the Azienda Ospedaliero-Universitaria Senese, COMEC, [N° 7/2021].

Teeth were sliced with a diamond saw at low speed. Slices were then treated with 40% phosphoric acid for 1 min, followed by 5.25% sodium hypochlorite for 10 min in an ultrasound bath.

The prepared dentin slices were placed in a 24-well plate and challenged with a suspension of *E. faecalis* PE bacteria (OD = 0.8), incubated at 37 °C for 1 h at 110 rpm. The supernatant was removed, and the slices were washed three times with PBS or a solution of the peptides in PBS at a concentration of 10 μM. Dentin slices were then incubated at 37 °C for 1 h in TSB. The supernatant suspension was serially diluted and plated in TSB-agar. CFU were counted after incubation overnight at 35 °C.

### 2.8. LPS-Peptide Binding Assay

M33D and M33Di/l were dissolved to working concentrations (10 μg/mL; 5 μg/mL; 1 μg/mL) in carbonate buffer (pH 9) and used to coat a 96-well ELISA strip plate. The negative controls were uncoated wells. The plate was sealed and incubated overnight at 4 °C. Each well was aspirated and washed three times with PBS + 0.05% Tween 20 and with PBS. The plate was blocked by adding 400 μL/well PBS and milk 3% and incubating for 2 h at 37 °C. LPS-Bio (Aurogene srl, Roma, Italy) was diluted in PBS and BSA 0.3% to a working concentration of 5 μg/mL. The negative control contained only PBS and BSA 0.3%. The plate was incubated in the dark at 30 °C for 30 min. After washing, 100 μL/well Streptavidin-POD (Sigma Aldrich, St. Louis, MO, USA) diluted 1:500 in PBS and milk 0.3% were added and incubated in the dark for 30 min at 30 °C. After washing, 150 μL/well of substrate solution was added, the reaction was stopped with 50 μL/well HCl 1M and the plate was read at 450 nm and 650 nm using a microplate spectrophotometer (Multiskan, Thermo Scientific, Waltham, MA, USA).

### 2.9. Macrophages Stimulation

RAW264.7 cells derived from murine macrophages (ATCC, Rockville, MD, USA) were grown in Dulbecco’s Modified Eagle’s Medium (DMEM) supplemented with glutamine, fetal bovine serum, and penicillin and streptomycin. In all experiments, cells were grown to 70–80% confluence. Cells were plated at 50.000 cells/well in a standard 96-well plate for 24 h at 37 °C and then were stimulated with LPS 2 μg/mL for 18 h in presence of the peptides at 16 μg/mL. After centrifugation, the supernatant of RAW264.7 was analyzed, and cytokines (IL-1β and TNF-α) were measured with a Milliplex kit (Merck-Millipore, Molsheim, France) following the manufacturer’s instruction.

### 2.10. Human Cardiac Fibroblast Modulation

Human cardiac fibroblasts (HCF) (Innoprot P104452) were grown in fibroblast basal medium containing fetal bovine serum, fibroblast growth supplement-2, and penicillin and streptomycin (Innoprot, Derio, Spain). Cells were plated in a standard 96-well plate at 500,000 cells/mL. After 24 h incubation at 37 °C, HCF were treated with LPS (2 μg/mL) and sequentially with the peptides (16 μg/mL) and left overnight at 37 °C. At the same time, a DuoSet ELISA plate (R&D Systems Inc., Minneapolis, MN, USA) was prepared according to the manufacturer’s instructions, adding 100 μL/well of the capture antibody. After 24 h, the DuoSet plate was washed and blocked with 1% BSA in PBS. The HCF supernatant was collected and centrifuged for 5 min at 1200 rpm and added to the DuoSet plate at 100 μL/well and incubated for 2 h at RT. After washing, the detection antibody was added and left for 2 h at RT. Finally, streptavidin-peroxidase was used for detection and the plates were read at 450 nm using a microplate spectrophotometer (Multiskan, Thermo Scientific). The IL6 concentration was calculated from a standard curve that was linearized by plotting the log of the six concentrations versus the log of the antibody.

### 2.11. Eukaryotic Cell Viability Assay

RAW 264.7 cells, 5 × 10^3^ per well, were seeded in 96-well plates and incubated for 24 h at 37 °C in a 5% CO_2_ atmosphere. A 100 μL volume of peptides at different concentrations in fresh medium was then added to the wells and kept for 24 h. Then 20 μL MTT (5 mg/mL) was added to each well, and the plate was incubated at 37 °C for 3 h. Finally, 120 μL HCl 4 mM in isopropanol was added to each well to dissolve the formazan crystals. Optical density was measured with a microplate reader (Bio-Rad, Hercules, CA, USA) at 570 nm. Cell viability was calculated by comparing the values of treated with those of untreated cells. Curves and EC50 were calculated using GraphPad Prism 5.03 software.

### 2.12. Hemolytic Activity

Whole human blood in EDTA was centrifuged (1100× *g*) for 10 min. The pellet of red blood cells was resuspended 1:100 in physiological solution (0.9% NaCl) and incubated with serial dilutions of the peptides from 1.25 to 340 µM, for 24 h at 37 °C. The supernatants were transferred into a 96-well plate, and the absorbance at 490 nm was measured with a microplate reader. Data for 100% hemolysis were obtained by adding 0.1% TritonX-100 in water to the cells. The saline solution was used as negative control.the The hemolysis rates of the peptides were calculated with the following equation: (%) = (A peptide − A physiological solution)/(A triton − A physiological solution) 100%; where A = absorbance.

### 2.13. Statistical Analysis

The data were plotted and analyzed using Excel (Microsoft, Redmond, WA, USA) and GraphPad Prism 5.0 software (GraphPad, San Diego, CA, USA), reported as means ± SD where relevant. Experiments were repeated at least twice.

## 3. Results

### 3.1. M33D and M33i/l

M33D and M33i/l are synthetic peptides built on a three-lysine core that allows four sequences on the same scaffold (Figure 1). The amino acids used for the sequences are all of the D-series, except the three lysines of the core. M33i/l carries a leucine in the place of the isoleucine of its parent compound M33D. Leucine, having one less stereocenter than isoleucine, gives fewer stereochemistry byproducts when transformed into a Fmoc-derivative and included in the amino acid sequence. It is therefore more affordable industrially for future development.

The tetrabranched scaffold already proved to increase the stability against proteases, compared to linear homologs [18]. The two peptides were incubated in serum for 72 h, and 31% of M33D and 25% of M33i/l were recovered intact in HPLC. The branched isomer in L configuration, M33L, and the linear L-homolog (KKIRVRLSA) were tested in the same assay for comparison. M33L showed similar stability to the two peptides M33D and M33i/l, but the linear analog was degraded completely within 4 h, as expected, considering that it carries all L-amino acids and that it is not branched (Figure 1C).

### 3.2. Minimal Inhibitory Concentration (MIC)

MICs of M33D and M33i/l were determined against strains of different Gram-negative and Gram-positive bacterial species (Table 1). The species were selected among the most relevant in medicine, as conveyors of resistance phenotypes and responsible for the majority of serious infections and include all pathogens of the ESKAPE group: *Enterococcus faecium, Staphylococcus aureus*, *Klebsiella pneumoniae*, *Acinetobacter baumannii*, *Pseudomonas aeruginosa*, and *Enterobacter* spp.

M33D was also tested against a *Corynebacterium*, two *Streptococci*, and three *Candida* strains.

Both M33D and M33i/l showed MIC against Gram-negative species with a range of activity between 1.5 and 6 μM for M33D and between 0.3 and 6 μM for M33i/l. Interestingly, M33i/l showed a very low MIC (0.3 μM) against the clinical isolate of *E. coli* LC711, which features resistance to colistin, a polypeptide antibiotic, used as a last-resort treatment for multidrug-resistant Gram-negative infections.

Regarding Gram-positive bacteria, M33D showed good activity against *Staphylococcus aureus* and *epidermidis*, ranging from 0.4 to 1.5 μM, including methicillin (*S. aureus* USA300, *S. aureus* 3851)- and vancomycin (*S. aureus* 3851)-resistant strains. *Enterococcus faecalis* and *faecium* were susceptible to M33D in the 0.8–3 μM range. *Corynebacterium*’s susceptibility to M33D was 1.5 μM. *Streptococcus* species *pneumoniae* and *pyogenes* were susceptible to M33D in the 6 μM range and *Candida* species in the 5–12 μM range.

*Pseudomonas aeruginosa*, *E. coli*, *K. pneumonia*, *Enterococcus*, and *Staphylococcus* species tested can be considered equally susceptible to M33i/l and M33D (Table 1). In all species tested, the MIC can be considered equal for the two peptides, since differences of one single dilution can be ascribed to experimental variability. The only exception is the *E. coli* strain LC711, featuring resistance to the polypeptide antibiotic colistin, where the i/l mutation in M33D changed the susceptibility from 1.5 to 0.3 μM. All the MICs are in line with previously reported well-known antimicrobial peptides [24].

### 3.3. Mechanism of Action of M33-D and M33i/l against Enterococcus faecalis

Most natural antibacterial peptides impair plasma membrane function, seriously challenging the bacteria cell. Membrane-targeting peptides can increase permeability to small ions or larger molecules and cause extensive membrane damage [25].

We visualized the kinetics of pore formation using membrane-impermeable fluorescent dyes such as propidium iodide (PI) and SYTOX green [26,27] in an *E. faecalis* post-endodontic treatment isolate (PE) and in *E. coli* LC711 clinical isolate [23], both oral colonizing species. Their fluorescence increases on binding to nucleic acids, which only happens when the cytoplasmic membrane is critically damaged.

Figure 2A–D shows that peptides M33D and M33i/l induced rapid permeabilization of the membrane after their addition at 2- and 4-fold the MIC. The results indicate that the two peptides can induce pore formation in the *E. faecalis* PE and *E. coli* LC711 membrane within 10–20 min.

A prediction of peptide tertiary structure was obtained with the computational protocol of APPTEST [19] using the linear analogs of M33D and M33i/l (Figure 2C,D). In both peptides, the sequence of amino acids takes a random configuration, i.e., linear and not helical; the top view of their structure shows an overall amphipathic structure, where the aliphatic nonpolar side chains of leucine, isoleucine, valine, and alanine arrange on opposite sides with respect to polar cationic lysines and arginines. The amphipathic characteristic underlies the embedding property of the peptide in the cell membrane, the first step in impairing membrane function that kills the bacteria.

### 3.4. Inhibition of Biofilm Formation

In biofilms, bacteria produce proteins and polysaccharides and grow as multicellular aggregates in an extracellular matrix that shelters their cells from environmental insults and host defenses. Biofilms are also more resistant to antimicrobial agents due to the physical barrier generated by the matrix and the state of dormancy of the bacteria [28,29,30]. M33D and M33i/l were tested for their antibiofilm activity against two species that belong to mouth microbiota, *E. faecalis* and *E. coli,* whose biofilm can threaten oral health. For both species, a reference strain and a clinical isolate were used as in vitro models. Microbial biofilms are regarded as a primary cause of periodontitis in teeth with infected root canals [31]. *E. faecalis* is the most prevalent species [32] in persistence intra-radicular infections, facilitated by its facultative anaerobic property. *E. faecalis* biofilm is particularly resistant to standard treatments and poses a major obstacle to endodontic disinfection of root canals.

Biofilm inhibition activity was evaluated on the basis of two parameters: the biofilm prevention concentration (BPC) and the minimal biofilm inhibition concentration (MBIC) [21]. The BPC is the lowest concentration of peptide that results in 10% lower biofilm formation compared to untreated controls. The biofilm was measured after removing planktonic cells by washing the wells, previously incubated for 24 h with the bacteria and the peptide. MBIC is the lowest concentration of peptide that results in a 10% reduction in preformed biofilm compared to untreated controls. A 24 h biofilm was challenged with serial dilutions of the peptides for 14 h. Then after removal of planktonic cells, the residual biofilm was measured. Crystal violet absorbance at 585 nm was used to measure biofilm in the case of both parameters.

As reported in Table 2, biofilm inhibition was tested against two strains of *E. faecalis*, ATCC 51,299 and PE endodontic isolate, and two strains of *E. coli*, TG1 reference strain and LC711 clinical isolate (Table 2). The BPC of M33D was equal to the MIC against *E. faecalis* reference strain ATCC 51,299 and clinical isolate PE. The BPC of M33i/l was double and equal to the MIC in *E. faecalis* reference strain and clinical isolate, respectively. The MBIC of both peptides against *E. faecalis* reference strain and PE was 6 μM, i.e., from two to three times the MIC. The results are particularly promising when compared to the MBICs of the two standard-of-care antibiotics against *E. faecalis*, i.e., ampicillin and linezolid, which are 2048 and 1024 times the MIC, respectively [33].

The BPC against the *E. coli* reference strain corresponded to the MIC for both peptides, whereas the MBIC was two times the MIC. *E. coli* LC711’s biofilm showed to be inhibited with a BPC of 1.5 μM and an MBIC of 3 μM for both peptides. M33i/l showed a notably low MIC against the clinical isolate of *E. coli* LC711, a strain that features resistance to colistin, an antimicrobial peptide used in the clinic, though the concentrations of M33i/l needed to disrupt the biofilm were in line with those against the reference strain.

### 3.5. Inhibition of Bacterial Regrowth on Dentin Slices by Washing with the Peptides

The standard operating procedure for the elimination of intracanal bacteria is irrigation of the root canal with antiseptic solutions [30]. The most common is sodium hypochlorite, though it often does not completely eradicate bacteria and may therefore be followed by relapse of infection. Dentin slices were used as a model of dental preparation before restoration to reduce bacterial bioburden and improve the outcome of bonded restorations [34]. Teeth slices were obtained from extracted human molars and premolars using a diamond saw. The dentin discs were sterilized, then infected and finally washed three times with a solution of the antimicrobial peptides at a concentration of 10 μM (Figure 3).

Washing with the two peptides reduced the bacterial burden to nearly one-twentieth with respect to control and to a lesser extent with respect to PBS wash, but still statistically significant (Figure 3). Irrigation, being a local treatment, allows use of a much higher concentration of M33D and M33i/l and also to alternate washes with antimicrobial peptides and hypochlorite solution, as would be feasible and realistic in a typical root canal treatment.

### 3.6. Immunomodulatory Activity of M33D and M33i/l

M33D was previously shown to neutralize lipopolysaccharide (LPS) and lipoteichoic acid (LTA) and also to consistently reduce expression of mediators of inflammation such as TNF-α, IL6, COX-2, KC, MIP-1, IP10, iNOS, and NF-kB [17]. This immunomodulatory activity is initiated by the peptides capturing and neutralizing LPS. As shown in Figure 4A,B, M33D and M33i/l could bind LPS-biotin with dose-dependent linearity (Figure 4A,B). LPS is taken as a benchmark in immunomodulatory models for its uncontroverted efficacy in triggering immune response in hematopoietic cells and fibroblasts.

RAW264.7 murine macrophages, when stimulated with LPS, release proinflammatory cytokines [17]. We tested the ability of the peptides to inhibit this effect with bead-based multiplex assays using the Luminex technology. RAW264.7 released 20% more IL-1β, compared to baseline, when stimulated for 18 h with LPS. M33D and M33i/l reduced IL-1β release by 16% and 14%, respectively (Figure 4C). TNF-α release from macrophages was greatly increased by LPS stimulation (+65%,) and M33D and M33i/l reduced it by 13% and 23%, respectively (Figure 4D). Peptides were used at 16 μg/mL corresponding to the MIC against *E. coli.*

This ability of the peptides to neutralize LPS also proved to underlie human cardiac fibroblast inhibition of IL6 release triggered by LPS.

Cardiac fibroblasts contribute to cardiac physiology with many functions, such as insulation of the conduction system and vascular maintenance [35]. They promote a response to insult in association with the immune system. Bacteria can trigger fibroblasts to produce inflammatory mediators (cytokines, chemokines, and growth factors) that recruit inflammatory cells from the circulation, amplifying the inflammatory process and inducing heart dysfunction through atherogenesis [36,37,38,39].

Human cardiac fibroblasts were stimulated for 24 h with LPS with or without M33D and M33Di/l peptides. The release of IL6, measured in ELISA, was reduced by both peptides: by 50% in the case of M33D and 35% in the case of M33Di/l (Figure 4E).

### 3.7. Cytotoxicity of M33D and M33i/l in Eukaryotic Cells

Cytotoxicity to eukaryotic cells was tested in RAW264.7 murine macrophages with 24 h incubation at 37 °C. M33D and M33i/l showed EC50s of 3.0 × 10^−5^ M and 1.8 × 10^−5^ M, respectively, both one order of magnitude higher than the average MIC in the different species (Figure 5A).

M33D and M33i/l did not show relevant hemolytic activity when incubated with red blood cells for 1 h at 37 °C and compared to triton at 1% in PBS (Figure 5B,C). Low toxicity of both peptides against eukaryotic cells confirms previous encouraging results obtained in vitro and in vivo with M33D [17].

## 4. Discussion

Bacteria of the human oral microbiota can occasionally turn into pathogens. Among these, Enterococci are major players, contributing to severe acute and chronic infections, both of which are atherogenic and lead to cardiovascular disease. M33D and M33i/l proved to be active against a panel of different Gram-positive and Gram-negative bacteria, including pathogens of the ESKAPE group, the most significant species in severe systemic infections. Microorganisms involved in periodontal and endodontal diseases, such as *E. coli*, *P. aeruginosa*, *Enterococci*, *Staphylococci*, *Streptococci*, and *Candida* showed acceptable susceptibility to M33D and M33i/l. The two peptides were active in the range of 0.4–6.0 μM, in all strains of all species, with an average of 1.5 μM, perfectly in line with other antimicrobial peptides.

M33D and M33i/l inhibited bacteria growth in planktonic form and also in biofilm. In fact, no difference occurred in the calculated 3D structures, in their interactions with membranes, and in antibiofilm activity. Indeed, M33i/l showed lower MIC against the *E. coli* strain LC711, featuring resistance to the polypeptide antibiotic colistin, with respect to M33D. The development of resistance by this strain against colistin, which has the same mechanism of action as M33D [25], could be accompanied by changes in the bacteria membrane and in its sensitivity that were addressed favorably by the i/l mutation of M33i/l, with respect to M33D. No other significant difference in the antibacterial activity of M33D and M33i/l was observed; active concentrations were equal or almost equal, indicating that the substitution of the isoleucine amino acid with leucine did not bring any change in the efficacy of the peptide and that the two structures can be considered almost equivalent.

Biofilm is a primary cause of periodontitis in teeth with infected root canals, and *E. faecalis* is the most prevalent species in persistence intra-radicular infections, facilitated by its facultative anaerobic property. Eradication of *E. faecalis* biofilm is almost impossible with standard antibiotics, but M33D and M33i/l showed good results in the standard in vitro model of biofilm prevention concentration (BPC) and minimal biofilm inhibition concentration (MBIC). Moreover, a dentin model of irrigation of the root canal was set up to test the efficacy of the two peptides. M33D and M33i/l reduced the bioburden of the teeth slices to nearly one-twentieth with respect to control by a simple wash with no incubation. The remarkable effect against *E. faecalis* biofilm and in the irrigation model can be ascribed to the fact that these amphipathic peptides have some detergent power due to the presence of charged and lipophilic residues on the same scaffold, which facilitate the disruption of biofilm. Moreover, the intended topical use of M33D and M33i/l in dentistry allows for high concentrations, also far beyond the cytotoxic EC50.

Many epidemiological studies suggest that periodontitis is associated with atherosclerotic cardiovascular disease: bacteria enter the systemic circulation, stimulating atherogenesis through endothelial damage and inflammation. Furthermore, patients with periodontitis constantly produce proinflammatory cytokines that can reach the liver and trigger release of acute-phase response proteins, causing endothelial damage and triggering atherogenesis. Importantly, M33D and M33i/l showed immunomodulatory activity, crucial for the healing of oral diseases, and especially for avoiding onset of atherosclerotic cardiovascular disease. The two peptides arrested the inflammatory cascade triggered by LPS in mouse macrophages and human heart fibroblasts. Thus, in the intended local use, M33D and M33i/l will likely kill bacteria by means of membrane functionality impairment, and then capture bacteria lipopolysaccharides preventing inflammation and atherosclerotic damage.

Antimicrobial peptides are already considered a good alternative to standard antibiotics to address dental infections [40], though a major impediment to the development of AMP as drugs is their short half-life due to protease susceptibility. M33D and M33i/l proved to be notably resistant to proteolytic degradation, thanks to their D configuration and also to their branched core, proving to be promising for the scope.

The use of antimicrobial peptides to irrigate tooth cavities and in medications is a new field of development that promises tools to fight dental infections and prevent their severe consequences, while at the same time protecting standard antibiotics against new outbreaks of antimicrobial resistance.

## Figures and Tables

**Figure 1 pharmaceutics-14-02081-f001:**
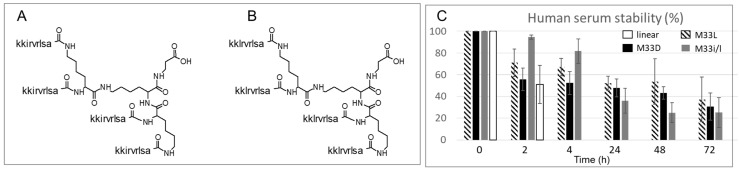
Structure of the branched antimicrobial peptides, M33D (**A**) and M33i/l (**B**). (**C**) Stability of the two tetrabranched peptides to serum proteolytic activity compared to a linear homolog, KKIRVRLSA, and to a branched analog in L configuration, M33L [18].

**Figure 2 pharmaceutics-14-02081-f002:**
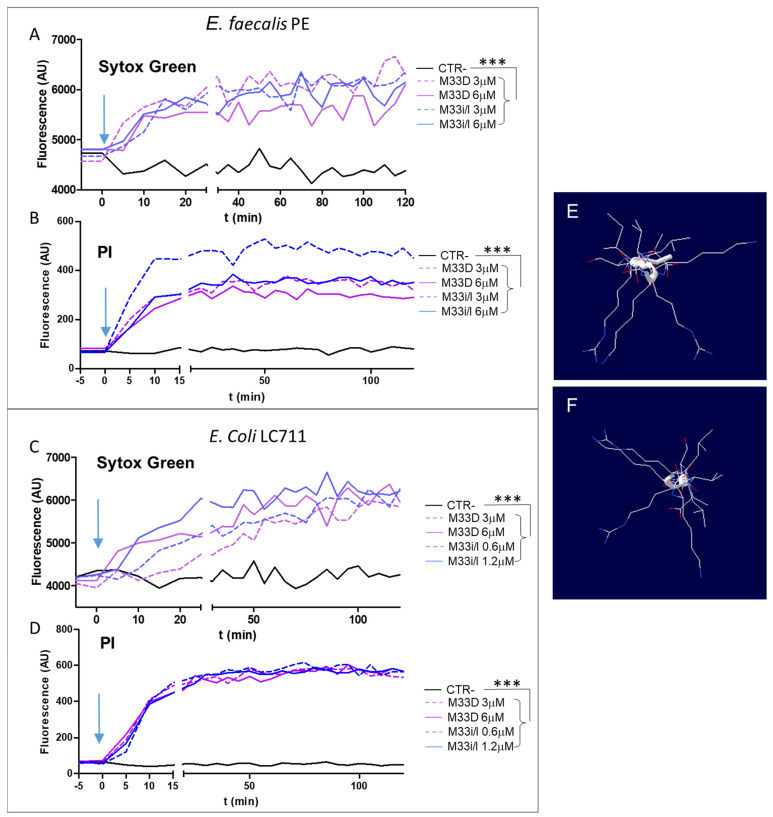
Observations on peptide mechanism of action. Membrane permeabilization (**A**) Sytox Green and (**B**) PI in *E. faecalis* PE and (**C**,**D**) in *E. coli* LC711 Experiments were performed with 10^8^ CFU/mL bacteria in PBS-glucose. The blue arrow indicates the time when the peptides were added. The peptides were used at 2- and 4-fold their MIC. *** *p* < 0.0001 for both peptides and both concentrations. De novo prediction of peptide tertiary structure by APPTEST for the linear analogs of M33D (**E**) and M33i/l (**F**).

**Figure 3 pharmaceutics-14-02081-f003:**
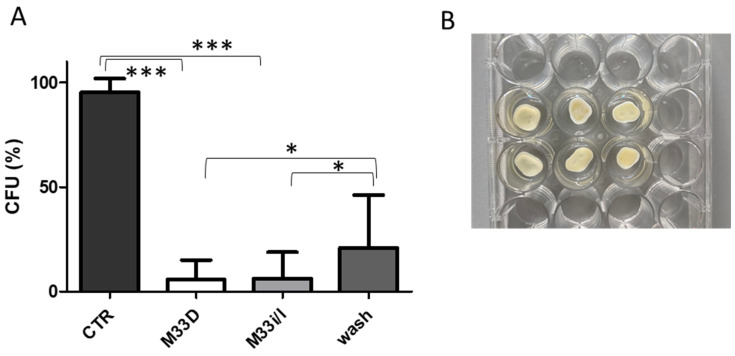
Inhibition of bacterial regrowth expressed in % CFU with respect to nonwashed control. (**A**) regrowth after washing dentin slices with peptides; *** *p* < 0.0001; * *p* < 0.0188 M33D; * *p* = 0.0423 M33i/l, CTR n = 5, M33D and M33i/l n = 20, washes n = 10; (**B**) image of dentine slices in a 24-well plate.

**Figure 4 pharmaceutics-14-02081-f004:**
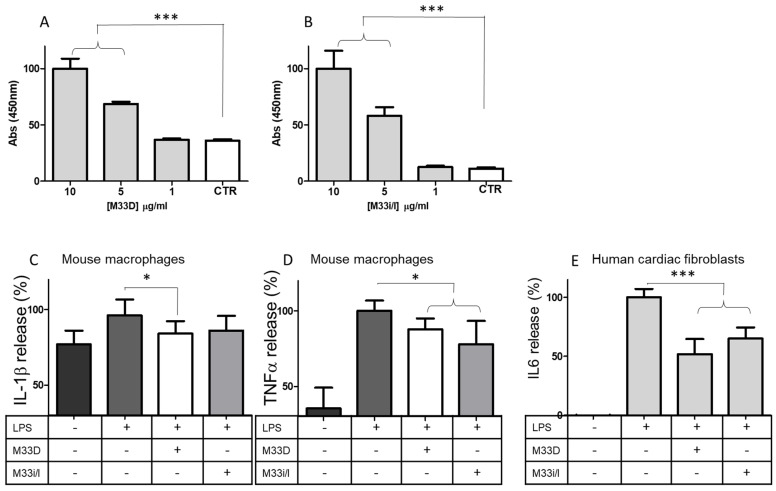
(**A**) M33D and (**B**) M33i/l binding to LPS (n = 6, for all groups, *** *p* < 0.0001). (**C**) IL 1β (n = 4 in the first group, 8 in the second, 6 in the third and fourth, * *p* = 0.0348) and (**D**) TNFα (n = 4 in the first and second group, 6 in the third and 4 in the fourth, * *p* < 0.05) release in LPS-stimulated RAW 264.7 macrophages in the presence of the peptides. (**E**) HCF release of IL6 in the presence of M33D and M33i/l (n = 3, *** *p* < 0.0001). Maximum IL6 release was 2.29 μM calculated from a linearized standard curve.

**Figure 5 pharmaceutics-14-02081-f005:**
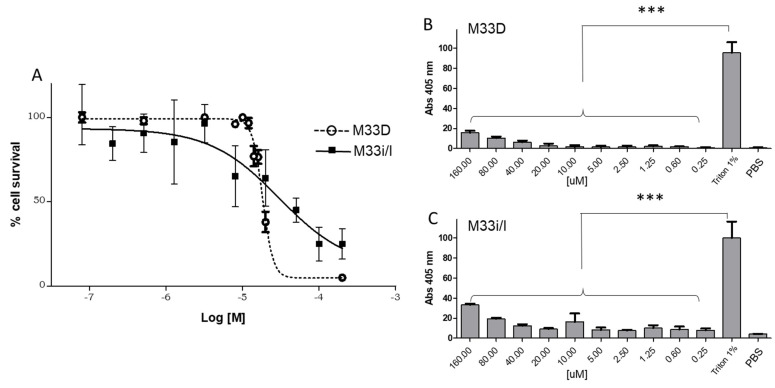
Cytotoxic and hemolytic effect of M33D and M33i/l. (**A**) RAW 264.7 were incubated with increasing concentrations of peptides for 24 h at 37 °C. Cell viability is reported as a percentage of the untreated cells (n = 3). Curves are obtained with GraphPad (nonlinear variable slope). (**B**) and (**C**) Hemolytic activity of M33D and M33i/l is reported as a percentage ± SD of 100% obtained with triton 1% in PBS after incubation for 1h at 37 °C [29], *** *p* < 0.0001 (n = 3).

**Table 2 pharmaceutics-14-02081-t002:** Biofilm inhibition. M33D and M33i/l biofilm inhibition against *E. faecalis ATCC 51299*, *E. faecalis* PE, *E. coli* TG1, and *E. coli* LC711. Values for minimal inhibitory concentration (MIC), biofilm prevention concentration (BPC), and minimal biofilm inhibitory concentration (MBIC) are reported.

	*E. faecalis* ATCC 51299		*E. faecalis* PE		*E. coli* TG1		*E. coli* LC711	
Peptide	M33D	M33i/l	M33D	M33i/l	M33D	M33i/l	M33D	M33i/l
**MIC** (μM)	3	1.5	1.5	1.5	1.5	1.5	1.5	0.3
**BPC** (μM)	3	3	1.5	1.5	1.5	1.5	1.5	1.5
**MBIC** (μM)	6	6	6	6	3	3	3	3

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
