# Peer review of "Antimicrobial Peptides Active in In Vitro Models of Endodontic Bacterial Infections Modulate Inflammation in Human Cardiac Fibroblasts"

_pharmaceutics, 2022, doi:10.3390/pharmaceutics14102081_

Round 1
Reviewer 1 Report
In this paper Marianantoni et al. reports antibacterial effects of two branched antimicrobial peptides (M33D & M33i/l) on a series of Gram-negative and Gram-positive bacterial species and strains relevant for the oral microbiota. Specifically, antibacterial (MIC) and bactericidal (Sytox/PI) and anti-biofilm activity, as well as some data on hemolytic activity and cytotoxicity. In addition, binding to LPS and reduction of LPS induced cytokine release is observed. Thus it is concluded, that these peptides may be useful for treating endodontic infections.
Overall the study is technically sound, but at a very early stage in terms of evaluating in vivo opportunities, and several issues must be addressed prior to considering publication.
Specific points:
1-It is very confusing and interferes with the reading of the paper and evaluation of the data that peptide concentration unit usage is not consequent (ug/ml and uM): uM should be used throughout.
2-The title is misleading and should be altered: This paper is not concerned with activity in bacterial infection, as no in vivo data are included.
3-Similar (identical?) cytotoxicity and hemolysis data for M33D (Fig. 5) have already been published by the authors (ref 17).
4-The cytotoxicity profiles for M33D and M33i/l are significantly different (fig. 5A). This must be explained/discussed.
5-It is stated that M33D “has long life” (not precise statement), according to ref 18. However, ref 18 is concerned with L-form peptides and not M33D.
6-M&M describes structure prediction, but no data/results are included relating to this. Should be removed.
7-It is not clear how fibroblast treatment with LPS and peptide was performed (line 222-4): Were they added simultaneously, mixed, …..?
8-The peptide concentration used for the fibroblast and the dentin experiments is also very close to cytotoxicity EC50!!!
9-Figure 3A and 3B represent identical data and one of them should be removed.
10-Figure 4A: Similar LPS binding data for M33D have already been published in ref 17.
11-In the legend to Fig4 C and D n is reported as 4-8 and 4-6, respectively. This is confusing. How were p-values calculated from these experiments?
12-A critical discussion of the results and their implications is totally missing.
Author Response
REVIEWER 1
Comments and Suggestions for Authors
In this paper Marianantoni et al. reports antibacterial effects of two branched antimicrobial peptides (M33D & M33i/l) on a series of Gram-negative and Gram-positive bacterial species and strains relevant for the oral microbiota. Specifically, antibacterial (MIC) and bactericidal (Sytox/PI) and anti-biofilm activity, as well as some data on hemolytic activity and cytotoxicity. In addition, binding to LPS and reduction of LPS induced cytokine release is observed. Thus it is concluded, that these peptides may be useful for treating endodontic infections.
Overall the study is technically sound, but at a very early stage in terms of evaluating in vivo opportunities, and several issues must be addressed prior to considering publication.
Response: We thank the Reviewer for the comments. We tried to address all the concerns raised and we hope that this extensively revised version of the Manuscript will meet the Journal’s requirements for publication.
Specific points:
REVIEWER 1-It is very confusing and interferes with the reading of the paper and evaluation of the data that peptide concentration unit usage is not consequent (ug/ml and uM): uM should be used throughout.
Authors. We thank the reviewer for the comment. We agree on the fact that the change in units can be misleading, so we turned everything into uM as suggested. Concerning the MIC, we actually added the mg/ml unit as it is a standard practice for microbiologists and so we provide it in brackets.
REVIEWER 1-The title is misleading and should be altered: This paper is not concerned with activity in bacterial infection, as no in vivo data are included.
Authors. Antibacterial activity is primarily tested in vitro in bacteria cell cultures, the definition of antibacterial or antimicrobial comes from the activity against microorganisms which is done in vitro. When testing new molecules in vivo, other observations are addressed, for example activity against a systemic infection, pharmacokinetic and toxicokinetic. In facts, we envisage a dental topical use of the peptides in the future, not a systemic use of it, we then setup an experimental model that reproduces as much as possible the treatment of endodontic bacterial infections. We better addressed the point in the new discussion paragraph.
REVIEWER 1-Similar (identical?) cytotoxicity and hemolysis data for M33D (Fig. 5) have already been published by the authors (ref 17).
Authors. Results of cytotoxicity against eukaryotic cells confirm the low toxicity of M33D observed in vitro and in vivo in ref 17. In this study, experiments were performed in parallel for the two peptides to have direct comparison and showed, as expected, that M33i/l is very similar to the parent peptide also in this aspect. We added a sentence in the paragraph that underlines that the result confirms previous ones.
REVIEWER 1-The cytotoxicity profiles for M33D and M33i/l are significantly different (fig. 5A). This must be explained/discussed.
Authors. Thank you for the observation. The two curves are not superimposable, but the calculated EC50 is almost equal for the two peptides. This is an experimental observation and the curve is drawn with Graphpad that calculates it using the data points provided. We added a sentence in the paragraph to comment the point and a reference regarding the method to calculate the curve in the figure legend.
REVIEWER 1-It is stated that M33D “has long life” (not precise statement), according to ref 18. However, ref 18 is concerned with L-form peptides and not M33D.
Authors. We thank the reviewer for the comment that allowed to add another result to the manuscript (protease stability).
REVIEWER 1-M&M describes structure prediction, but no data/results are included relating to this. Should be removed.
Authors. In paragraph 3.3, there’s actually the description of the prediction of peptides’ tertiary structure obtained with the computational protocol of APPTEST. The result is also shown in the Figure 2C and D.
REVIEWER 1-It is not clear how fibroblast treatment with LPS and peptide was performed (line 222-4): Were they added simultaneously, mixed, …..?
Authors. LPS and peptides were added one after the other. The sentence was modified following the suggestion.
REVIEWER 1-The peptide concentration used for the fibroblast and the dentin experiments is also very close to cytotoxicity EC50!!!
Authors. Yes, that is perfectly right and it is now discussed in the discussion session in two points. 1) M33D and M33i/l reduced the bioburden of the teeth slices to nearly one twentieth with respect to control, by a simple wash, with no incubation. The remarkable effect against E. faecalis biofilm and in the irrigation model, can be ascribed to the fact that these amphipathic peptides have some detergent power due to the presence of charged and lipophilic residues on the same scaffold, that facilitate the disruption of biofilm. The intended topical use of M33D and M33i/l in dentistry allows for high concentrations, also far beyond the cytotoxic EC50. 2) Fibroblasts experiment. The two peptides arrested the inflammatory cascade triggered by LPS in mouse macrophages and human heart fibroblasts. Thus, in the intended local use, M33D and M33i/l will likely kill bacteria by means of membrane functionality impairment, and then capture bacteria lipopolysaccharides preventing inflammation. Again the local use allows for high concentrations.
REVIEWER 1-Figure 3A and 3B represent identical data and one of them should be removed.
Authors. Panel B was removed as suggested, we shown now only the %CFU.
REVIEWER 1-Figure 4A: Similar LPS binding data for M33D have already been published in ref 17.
Authors. No LPS binding is shown in ref 17, the paper reports anti-inflammatory activity in vivo. The experiment reported in Figure 4A of this manuscript is new and never published before.
REVIEWER 1-In the legend to Fig4 C and D n is reported as 4-8 and 4-6, respectively. This is confusing. How were p-values calculated from these experiments?
Authors. The legend has been modified according to the suggestion.
REVIEWER 1-A critical discussion of the results and their implications is totally missing.
Authors. The discussion was rewritten.
Reviewer 2 Report
This manuscript, entitled “Antimicrobial peptides active against endodontic bacterial infections modulate inflammation in human cardiac fibroblasts”, aimed at identifying the efficacy of M33D and of its analogue M33i/l as antibiotics for local use in dentistry. The identification and development of novel antimicrobial peptides are of importance. However, due to some drawbacks, my suggestion is major revision.
1. The structure and characteristics of the two antimicrobial peptides are of importance. However, the information described in the manuscript is limited. In 3.1, only a few sentences were presented. Please expand.
2. The strains included in the MIC test are listed in Table1. What’s the principle and standard of selecting these strains? It’s relatively representative to include Gram-negative and Gram-positive bacterial species, as well as fungi. However, why some other important pathogens including Acinetobacter baumannii, Enterococcus faecium were not included. Also, why M33Di/l was not tested against a Corynebacterium, two Streptococci and three Candida strains?
3. Considering the mechanism of action of the two peptides, why only Enterococcus faecalis was used? At least one each of Gram-negative and Gram-positive bacterial species, and fungi would be interested to be included.
4. In biofilm formation assay, two E. faecalis strains and one E. coli strain were included. It’s confusing about the rationale and standard on the variety in usage of strains in each part.
5. In the rest experiments, was only E. faecalis included? If so, I suggest the authors either change the title to specifically indicate E. faecalis (which would reduce the significance), or include the tests on more species (more preference).
6. The current Discussion part is rather a Conclusion. Please expand discussion.
Author Response
REVIEWER 2 This manuscript, entitled “Antimicrobial peptides active against endodontic bacterial infections modulate inflammation in human cardiac fibroblasts”, aimed at identifying the efficacy of M33D and of its analogue M33i/l as antibiotics for local use in dentistry. The identification and development of novel antimicrobial peptides are of importance. However, due to some drawbacks, my suggestion is major revision.
Authors. We thank the Reviewer for the comments. We tried to address all the concerns raised and we hope that this extensively revised version of the Manuscript will meet the Journal’s requirements for publication.
REVIEWER 2. The structure and characteristics of the two antimicrobial peptides are of importance. However, the information described in the manuscript is limited. In 3.1, only a few sentences were presented. Please expand.
Authors. Paragraph 3.1 was improved giving significance to the branched structure with a stability test. The 3D structure is also better described in 3.3.
REVIEWER 2. The strains included in the MIC test are listed in Table1. What’s the principle and standard of selecting these strains? It’s relatively representative to include Gram-negative and Gram-positive bacterial species, as well as fungi. However, why some other important pathogens including Acinetobacter baumannii, Enterococcus faecium were not included.
Authors. Thank you for the observation. We modified the paragraph concerning the MIC, giving reasons of the choices. We also added the data on Acinetobacter baumannii and Enterococcus faecium results, citing a former study.
REVIEWER 2. Also, why M33Di/l was not tested against a Corynebacterium, two Streptococci and three Candida strains?
Authors. The tests on the two peptides were performed in different times and it was impossible to test M33i/l against the strains suggested. We added a few sentences commenting the fact that M33i/l is almost completely equal in activity in all species and strains, in biofilm inhibition and in membrane disruption, making it very likely completely interchangeable with the parent peptide M33D.
REVIEWER 2. Considering the mechanism of action of the two peptides, why only Enterococcus faecalis was used? At least one each of Gram-negative and Gram-positive bacterial species, and fungi would be interested to be included.
Authors. We added a new experiment with Sytox green and PI in a clinical isolate of E.coli. We chose these two species because more relevant for oral infections. We think that the study of different mechanism of action in Gram-negative, Gram-positive and fungi is very interesting and deserves a completely dedicated study, beyond the premises of this manuscript that focuses on the development of two new branched peptides in endodontic treatments, to fight dental infections and prevent their severe consequences, while at the same time protecting standard antibiotics against new outbreaks of antimicrobial resistance.
REVIEWER 2. In biofilm formation assay, two E. faecalis strains and one E. coli strain were included. It’s confusing about the rationale and standard on the variety in usage of strains in each part.
Authors. We performed and added a new experiment of biofilm inhibition using a clinical isolate of E.coli. Now we have two reference strains and two clinical isolates of bacteria related to endodontic infections and we hope now it will be considered more balanced.
REVIEWER 2. In the rest experiments, was only E. faecalis included? If so, I suggest the authors either change the title to specifically indicate E. faecalis (which would reduce the significance), or include the tests on more species (more preference).
Authors. We focused on E. faecalis since it is a strain of endodontic interest and is considered extraordinarily difficult to be eradicated, due to its extremely resistant biofilm. As the results on biofilm show, the activity of the two peptides against E. f. are notable and deserve future development, that is the scope of the study. Though, we added two extra experiments, mechanism of action and biofilm eradication with another strain, to address your concern.
- The current Discussion part is rather a Conclusion. Please expand discussion.
Authors. Thank you for the comment, the discussion was rewritten.
Reviewer 3 Report
The MS described the antimicrobial and antibiofilm activity of the peptides with immunomodulatory effect
Result: table (2) title need to be describes in more detail
Fig 3; A is representing CFU or CFU/ml
Also B; line gride need to be removed
Fig 5; Significance marks need to be added
The data need to be discussed as the discussion is very consized.
Author Response
REVIEWER 3
immunomodulatory effect
Result: table (2) title need to be describes in more detail
Authors. Thank you for the comment, we modified the title of the Table.
REVIEWER 3 Fig 3; A is representing CFU or CFU/ml.
Authors. CFU.
REVIEWER 3 Also B; line gride need to be removed
Authors. Panel B was removed as considered redundant by other reviewers.
REVIEWER 3 Fig 5; Significance marks need to be added
Authors. Thank you, corrected.
REVIEWER 3 The data need to be discussed as the discussion is very consized.
Authors. Thank you for the comment, the discussion was rewritten.
Round 2
Reviewer 1 Report
Two issues remain:
The title is misleading and must be changed. The study is not addressing ”activity against endodontic bacterial infection”, because this would include infections in the dental pulp, and would imply a medical dentistry situation (in vivo). The study is demonstrating antibacterial activity using a dental in vitro (ex vivo) model.
The stability discussion is not valid (lines 289-292: The tetrabranched scaffold increased stability against proteases, compared to a linear homologue (KKIRVRLSA). The authors are comparing an L-form of the peptide with branched D-form, and the stabilization will be mainly due to the choice of the non-biological L-form of the peptide and not to the scaffold.
Author Response
Reviewer: The title is misleading and must be changed. The study is not addressing ”activity against endodontic bacterial infection”, because this would include infections in the dental pulp, and would imply a medical dentistry situation (in vivo). The study is demonstrating antibacterial activity using a dental in vitro (ex vivo) model.
Authors: Thank you for the suggestion. We modified the title in order to point out we worked on an in vitro dental model
Reviewer: The stability discussion is not valid (lines 289-292: The tetrabranched scaffold increased stability against proteases, compared to a linear homologue (KKIRVRLSA). The authors are comparing an L-form of the peptide with branched D-form, and the stabilization will be mainly due to the choice of the non-biological L-form of the peptide and not to the scaffold.
Authors: Thank you, we modified the discussion regarding the stability. We have had many previous assessments on stability of peptides determined by their branched structure, that we list below, but which do not fit in this discussion, so we omit to mention them in the manuscript. Nevertheless, we agree with your comment and we adjusted the results and discussion, mentioning the proteolytic resistance brought by the D-form. We also added the results of the stability of the branched M33L isomer to the table, so that the contribution of the branched core to the stability is more evident.
Falciani, C., Lozzi, L., Pini, A., Corti, F., Fabbrini, M., Bernini, A., Lelli, B., Niccolai, N., & Bracci, L. (2007). Molecular basis of branched peptides resistance to enzyme proteolysis. Chemical biology & drug design, 69(3), 216–221. https://doi.org/10.1111/j.1747-0285.2007.00487.x
Pini, A., Falciani, C., & Bracci, L. (2008). Branched peptides as therapeutics. Current protein & peptide science, 9(5), 468–477. https://doi.org/10.2174/138920308785915227
Reviewer 2 Report
The revision of this manuscript has been improved, however, a few points still need to be addressed.
1. The authors have added the test on E. coli for a few experiments, but E. faecalis and E. coli strains are not representative enough as the cause of endodontic bacterial infections modulate inflammation in human cardiac fibroblasts. I suggest the authors modify to title to fit more on the contents.
2. The current Discussion part has been expanded. However, it’s like a concise repeat of the results. More discussion and comparison based on the previous reported studies should be included.
Author Response
Reviewer: The authors have added the test on E. coli for a few experiments, but E. faecalis and E. coli strains are not representative enough as the cause of endodontic bacterial infections modulate inflammation in human cardiac fibroblasts. I suggest the authors modify to title to fit more on the contents.
Authors: We modified the title in order to point out we worked on an in vitro model of possible oral infections and consequent inflammation.
Reviewer: The current discussion part has been expanded. However, it’s like a concise repeat of the results. More discussion and comparison based on the previous reported studies should be included.
Authors: Thank you for the suggestion. We further expanded the discussion and included one more very recent reference.
Reviewer 3 Report
In the present study, M33D and M33i/l are long-life peptides showed antimicrobial activity against Gram-negative and Gram-positive bacteria. It has immunomodulatory activity, crucial for the healing of oral diseases, and especially for avoiding onset of atherosclerotic cardiovascular disease and low toxicity to mammalian cells.
The MS could be accepted in the formate
Author Response
Thank you for the favourable comment.